# HOUDINI: Lifelong Learning as Program Synthesis

**Lazar Valkov**
University of Edinburgh
L.Valkov@sms.ed.ac.uk

**Dipak Chaudhari**
Rice University
dipakc@rice.edu

**Akash Srivastava**
University of Edinburgh
Akash.Srivastava@ed.ac.uk

**Charles Sutton**
University of Edinburgh,
The Alan Turing Institute, and Google Brain
charlessutton@google.com

**Swarat Chaudhuri**
Rice University
swarat@rice.edu

## Abstract

We present a *neurosymbolic framework* for the lifelong learning of algorithmic tasks that mix perception and procedural reasoning. Reusing high-level concepts across domains and learning complex procedures are key challenges in lifelong learning. We show that a *program synthesis* approach that combines gradient descent with combinatorial search over programs can be a more effective response to these challenges than purely neural methods. Our framework, called HOUDINI, represents neural networks as strongly typed, differentiable functional programs that use symbolic higher-order combinators to compose a library of neural functions. Our learning algorithm consists of: (1) a symbolic program synthesizer that performs a type-directed search over parameterized programs, and decides on the library functions to reuse, and the architectures to combine them, while learning a sequence of tasks; and (2) a neural module that trains these programs using stochastic gradient descent. We evaluate HOUDINI on three benchmarks that combine perception with the algorithmic tasks of counting, summing, and shortest-path computation. Our experiments show that HOUDINI transfers high-level concepts more effectively than traditional transfer learning and progressive neural networks, and that the typed representation of networks significantly accelerates the search.

## 1 Introduction

*Differentiable programming languages* [25, 29, 8, 15, 10, 39, 34] have recently emerged as a powerful approach to the task of engineering deep learning systems. These languages are syntactically similar to, and often direct extensions of, traditional programming languages. However, programs in these languages are differentiable in their parameters, permitting gradient-based parameter learning. The framework of differentiable languages has proven especially powerful for representing architectures that have input-dependent structure, such as deep networks over trees [35, 2] and graphs [23, 19].

A recent paper by Gaunt et al. [14] points to another key appeal of high-level differentiable languages: they facilitate *transfer* across learning tasks. The paper gives a language called NEURAL TERPRET (NTPT) in which programs can invoke a library of trainable neural networks as subroutines. The parameters of these "library functions" are learned along with the code that calls them. The modularity of the language allows knowledge transfer, as a library function trained on a task can be reused in later tasks. In contrast to standard methods for transfer learning in deep networks, which re-use the first few layers of the network, neural libraries have the potential to enable reuse of higher, more abstract levels of the network, in what could be called *high-level transfer*. In spite of its promise, though, inferring the structure of differentiable programs is a fundamentally hard problem. While

NTPT and its predecessor TERPRET [15] allow some aspects of the program structure to be induced, a detailed hand-written template of the program is required for even the simplest tasks.

In this paper, we show that algorithmic ideas from *program synthesis* can help overcome this difficulty. The goal in program synthesis [3, 36, 13] is to discover programs (represented as terms following a specified syntax) that accomplish a given task. Many symbolic algorithms for the problem have been proposed in the recent past [16]. These algorithms can often outperform purely neural approaches on procedural tasks [15]. A key insight behind our approach is that these methods naturally complement stochastic gradient descent (SGD) in the induction of differentiable programs: while SGD is an effective way of learning program parameters, each step in a symbolic search can explore large changes to the program structure.

A second feature of our work is a representation of programs in a *typed functional language*. Such a representation is natural because functional combinators can compactly describe many common neural architectures [26]. For example, `fold` combinators can describe recurrent neural networks, and convolution over data structures such as lists and graphs can also be naturally expressed as functional combinators. Such representations also facilitate search, as they tend to be more canonical, and as the type system can help prune infeasible programs early on in the search [13, 27].

Concretely, we present a *neurosymbolic* learning framework, called HOUDINI, which is to our knowledge the first method to use symbolic methods to induce differentiable programs. In HOUDINI, a program synthesizer is used to search over networks described as strongly typed functional programs, whose parameters are then tuned end-to-end using gradient descent. Programs in HOUDINI specify the architecture of the network, by using functional combinators to express the network's connections, and can also facilitate learning transfer, by letting the synthesizer choose among library functions. HOUDINI is flexible about how the program synthesizer is implemented: here, we use and compare two implementations, one performing top-down, type-directed enumeration with a preference for simpler programs, and the other based on a type-directed evolutionary algorithm. The implementation for HOUDINI is available online [1].

We evaluate HOUDINI in the setting of *lifelong learning* [38], in which a model is trained on a series of tasks, and each training round is expected to benefit from previous rounds of learning. Two challenges in lifelong learning are *catastrophic forgetting*, in which later tasks overwrite what has been learned from earlier tasks, and *negative transfer*, in which attempting to use information from earlier tasks hurts performance on later tasks. Our use of a neural library avoids both problems, as the library allows us to freeze and selectively re-use portions of networks that have been successful.

Our experimental benchmarks combine perception with algorithmic tasks such as counting, summing, and shortest-path computation. Our programming language allows parsimonious representation for such tasks, as the combinators used to describe network structure can also be used to compactly express rich algorithmic operations. Our experiments show that HOUDINI can learn nontrivial programs for these tasks. For example, on a task of computing least-cost paths in a grid of images, HOUDINI discovers an algorithm that has the structure of the Bellman-Ford shortest path algorithm [7], but uses a learned neural function that approximates the algorithm's "relaxation" step. Second, our results indicate that the modular representation used in HOUDINI allows it to transfer high-level concepts and avoid negative transfer. We demonstrate that HOUDINI offers greater transfer than progressive neural networks [32] and traditional "low-level" transfer [40], in which early network layers are inherited from previous tasks. Third, we show that the use of a higher-level, typed language is critical to scaling the search for programs.

The contributions of this paper are threefold. First, we propose the use of symbolic program synthesis in transfer and lifelong learning. Second, we introduce a specific representation of neural networks as typed functional programs, whose types contain rich information such as tensor dimensions, and show how to leverage this representation in program synthesis. Third, we show that the modularity inherent in typed functional programming allows the method to transfer high-level modules, to avoid negative transfer and to achieve high performance with a small number of examples.

**Related Work.**  HOUDINI builds on a known insight from program synthesis [16]: that functional representations and type-based pruning can be used to accelerate search over programs [13, 27, 20]. However, most prior efforts on program synthesis are purely symbolic and driven by the Boolean goals. HOUDINI repurposes these methods for an optimization setting, coupling them with gradient-based learning. A few recent approaches to program synthesis do combine symbolic and neural

**Types** $\tau$:

$$
\begin{array}{lcl}
\tau & ::= & Atom \mid ADT \mid F \\
TT & ::= & \texttt{Tensor}\langle Atom\rangle[m_1][m_2]\ldots[m_k] \\
F & ::= & ADT \mid F_1 \to F_2
\end{array}
\qquad
\begin{array}{lcl}
Atom & ::= & \texttt{bool} \mid \texttt{real} \\
ADT & ::= & TT \mid \alpha\langle TT\rangle \\
& & .
\end{array}
$$

**Programs e over library** $\mathcal{L}$: $\quad e \quad ::= \oplus_w : \tau_0 \mid e_0 \circ e_1 \mid \mathbf{map}_\alpha\ e \mid \mathbf{fold}_\alpha\ e \mid \mathbf{conv}_\alpha\ e.$

Figure 1: Syntax of the HOUDINI language. Here, $\alpha$ is an ADT, e.g., $\texttt{list}$; $m_1,\ldots,m_k \geq 1$ define the shape of a tensor; $F_1 \to F_2$ is a function type; $\oplus_w \in \mathcal{L}$ is a neural library function that has type $\tau_0$ and parameters $w$; and $\circ$ is the composition operator. $\mathbf{map}$, $\mathbf{fold}$, and $\mathbf{conv}$ are defined below.

methods [11, 6, 12, 28, 18]. Unlike our work, these methods do not synthesize differentiable programs. The only exception is NTPT [14], which neither considers a functional language nor a neurosymbolic search. Another recent method that creates a neural library is progress-and-compress [33], but this method is restricted to feedforward networks and low-level transfer. It is based on progressive networks [32], a method for lifelong learning based on low-level transfer, in which lateral connections are added between the networks for the all the previous tasks and the new task.

*Neural module networks* (NMNs) [4, 17] select and arrange reusable neural modules into a program-like network, for visual question answering. The major difference from our work is that NMNs require a natural language input to guide decisions about which modules to combine. HOUDINI works without this additional supervision. Also, HOUDINI can be seen to perform a form of *neural architecture search*. Such search has been studied extensively, using approaches such as reinforcement learning, evolutionary computation, and best-first search [42, 24, 31, 41]. HOUDINI operates at a higher level of abstraction, combining entire networks that have been trained previously, rather than optimizing over lower-level decisions such as the width of convolutional filters, the details of the gating mechanism, and so on. HOUDINI is distinct in its use of functional programming to represent architectures compactly and abstractly, and in its extensive use of types in accelerating search.

## 2 The HOUDINI Programming Language

HOUDINI consists of two components. The first is a typed, higher-order, functional language of differentiable programs. The second is a learning procedure split into a symbolic module and a neural module. Given a task specified by a set of training examples, the symbolic module enumerates parameterized programs in the HOUDINI language. The neural module uses gradient descent to find optimal parameters for synthesized programs; it also assesses the quality of solutions and decides whether an adequately performant solution has been discovered.

The design of the language is based on three ideas:

- The ubiquitous use of *function composition* to glue together different networks.
- The use of *higher-order combinators* such as **map** and **fold** to uniformly represent neural architectures as well as patterns of recursion in procedural tasks.
- The use of a strong *type discipline* to distinguish between neural computations over different forms of data, and to avoid generating provably incorrect programs during symbolic exploration.

Figure 1 shows the grammar for the HOUDINI language. Here, $\tau$ denotes types and $e$ denotes programs. Now we elaborate on the various language constructs.

**Types.** The "atomic" data types in HOUDINI are booleans ($\texttt{bool}$) and reals. For us, $\texttt{bool}$ is relaxed into a real value in $[0, 1]$, which for example, allows the type system to track if a vector has been passed through a sigmoid. *Tensors* over these types are also permitted. We have a distinct type $\texttt{Tensor}\langle Atom\rangle[m_1][m_2]\ldots[m_k]$ for tensors of shape $m_1 \times \cdots \times m_k$ whose elements have atomic type $Atom$. (The dimensions $m_1,\ldots,m_k$, as well as $k$ itself, are bounded to keep the set of types finite.) We also have function types $F_1 \to F_2$, and abstract data types (ADTs) $\alpha\langle TT\rangle$ parameterized by a tensor type $TT$. Our current implementation supports two kinds of ADTs: $\texttt{list}\langle TT\rangle$, lists with elements of tensor type $TT$, and $\texttt{graph}\langle TT\rangle$, graphs whose nodes have values of tensor type $TT$.

**Programs.** The fundamental operation in HOUDINI is *function composition*. A composition operation can involve functions $\oplus_w$, parameterized by weights $w$ and implemented by neural networks, drawn from a library $\mathcal{L}$. It can also involve a set of symbolic higher-order combinators that are

guaranteed to preserve end-to-end differentiability and used to implement high-level network architectures. Specifically, we allow the following three families of combinators. The first two are standard constructs for functional languages, whereas the third is introduced specifically for deep models.

- Map combinators $\mathbf{map}_{\alpha\langle\tau\rangle}$, for ADTs of the form $\alpha\langle\tau\rangle$. Suppose $e$ is a function. The expression $\mathbf{map}_{\mathtt{list}\langle\tau\rangle}\ e$ is a function that, given a list $[a_1,\ldots,a_k]$, returns the list $[e(a_1),\ldots,e(a_k)]$. The expression $\mathbf{map}_{\mathtt{graph}_\tau}\ e$ is a function that, given a graph $G$ whose $i$-th node is labeled with a value $a_i$, returns a graph that is identical to $G$, but whose $i$-th node is labeled by $e(a_i)$.
- Left-fold combinators $\mathbf{fold}_{\alpha\langle\tau\rangle}$. For a function $e$ and a term $z$, $\mathbf{fold}_{\mathtt{list}\langle\tau\rangle}\ e\ z$ is the function that, given a list $[a_1,\ldots,a_k]$, returns the value $(e\ (e\ \ldots(e\ (e\ z\ a_1)\ a_2)\ldots)\ a_k)$. To define fold over a graph, we assume a linear order on the graph's nodes. Given $G$, the function $\mathbf{fold}_{\mathtt{graph}\langle\tau\rangle}\ e\ z$ returns the fold over the list $[a_1,\ldots,a_k]$, where $a_i$ is the value at the $i$-th node in this order.
- Convolution combinators $\mathbf{conv}_{\alpha\langle\tau\rangle}$. Let $p>0$ be a fixed constant. For a "kernel" function $e$, $\mathbf{conv}_{\mathtt{list}\langle\tau\rangle}\ e$ is the function that, given a list $[a_1,\ldots,a_k]$, returns the list $[a'_1,\ldots,a'_k]$, where $a'_i = e\ [a_{i-p},\ldots,a_i,\ldots,a_{i+p}]$. (We define $a_j = a_1$ if $j<1$, and $a_j = a_k$ if $j>k$.) Given a graph $G$, the function $\mathbf{conv}_{\mathtt{graph}\langle\tau\rangle}\ e$ returns the graph $G'$ whose node $u$ contains the value $e\ [a_{i_1},\ldots,a_{i_m}]$, where $a_{i_j}$ is the value stored in the $j$-th neighbor of $u$.

Every neural library function is assumed to be annotated with a type. Using programming language techniques [30], HOUDINI assigns a type to each program $e$ whose subexpressions use types consistently (see supplementary material). If it is impossible to assign a type to $e$, then $e$ is *type-inconsistent*. Note that complete HOUDINI programs do not have explicit variable names. Thus, HOUDINI follows the *point-free* style of functional programming [5]. This style permits highly succinct representations of complex computations, which reduces the amount of enumeration needed during synthesis.

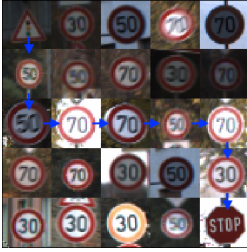

Figure 2: A grid of images from the GTSRB dataset [37]. The least-cost path from the top left to the bottom right node is marked.

**HOUDINI for deep learning.** The language has several properties that are useful for specifying deep models. First, any complete HOUDINI program $e$ is differentiable in the parameters $w$ of the neural library functions used in $e$. Second, common deep architectures can be compactly represented in our language. For example, deep feedforward networks can be represented by $\oplus_1 \circ \cdots \circ \oplus_k$, where each $\oplus_i$ is a neural function, and recurrent nets can be expressed as $\mathbf{fold}_{\mathtt{list}\langle\tau\rangle}\ \oplus\ z$, where $\oplus$ is a neural function and $z$ is the initial state. Graph convolutional networks can be expressed as $\mathbf{conv}_{\mathtt{graph}\langle\tau\rangle}\ \oplus$. Going further, the language can be easily extended to handle bidirectional recurrent networks, attention mechanisms, and so on.

**Example: Shortest path in a grid of images.** To show how HOUDINI can model tasks that mix perception and procedural reasoning, we use an example that generalizes the navigation task of Gaunt et al. [14]. Suppose we are given a grid of images (e.g., Figure 2), whose elements represent speed limits and are connected horizontally and vertically, but not diagonally. Passing through each node induces a penalty, which depends on the node's speed limit, with lower speed limits having a higher penalty. The task is to predict the minimum cost $d(u)$ incurred while traveling from a fixed starting point $init$ to every other node $u$.

One way to compute these costs is using the Bellman-Ford shortest-path algorithm [7], whose $i$-th iteration computes an estimated minimum cost $d_i(u)$ of travel to each node $u$ in the graph. The cost estimates for the $(i+1)$-th iteration are computed using a *relaxation* operation: $d_{i+1}(u) := \min(d_i(u), \min_{v \in Adj(u)} d_i(v) + w(v))$, where $w(v)$ is the penalty and $Adj(u)$ the neighbors of $u$. As the update to $d_i(u)$ only depends on values at $u$ and its neighbors, the relaxation step can be represented as a graph convolution. As described in Section 4, HOUDINI is able to discover an approximation of this program purely from data. The synthesized program uses a graph convolution, a graph map, a neural module that processes the images of speed limits, and a neural module that approximates the relaxation function.

## 3 Learning Algorithm

Now we define our learning problem. For a HOUDINI program $e_w$ parameterized by a vector $w$, let $e[w \mapsto v]$ be the function for the specific parameter vector $v$, i.e. by substituting $w$ by $v$ in $e$.

Suppose we have a library $\mathcal{L}$ of neural functions and a training set $\mathcal{D}$. As usual, we assume that $\mathcal{D}$ consists of i.i.d. samples from a distribution $p_{data}$. We assume that $\mathcal{D}$ is properly typed, i.e., every training instance $(x_i, y_i) \in \mathcal{D}$ has the same type, which is known. This means that we also know the type $\tau$ of our target function. The goal in our learning problem is to discover a program $e_w^*$ of type $\tau$, and values $v$ for $w$ such that $e_w^*[w \mapsto v] = \mathrm{argmin}_{e \in Progs(\mathcal{L}), w \in \mathbb{R}^n}(\mathbf{E}_{x \sim p_{data}}[l(e, \mathcal{D}, x)])$, where $Progs(\mathcal{L})$ is the universe of all programs over $\mathcal{L}$, and $l$ is a suitable loss function.

Our algorithm for this task consists of a symbolic program synthesis module called GENERATE and a gradient-based optimization module called TUNE. GENERATE repeatedly generates parameterized programs $e_w$ and "proposes" them to TUNE. TUNE uses stochastic gradient descent to find parameter values $v$ for $e_w$ that lead to the optimal value of the loss function on a training set, and produces a program $e = e_w[w \mapsto v]$ with instantiated parameters. The final output of the algorithm is a program $e^*$, among all programs $e$ as above, that leads to optimal loss on a validation set.

As each program proposed by GENERATE is subjected to training, GENERATE can only afford to propose a small number of programs, out of the vast combinatorial space of all programs. Selecting these programs is a difficult challenge. We use and compare two strategies for this task. Now we sketch these strategies; for more details, see the supplementary material.

- The first strategy is *top-down iterative refinement*, similar to the algorithm in the $\lambda^2$ program synthesizer [13]. Here, the synthesis procedure iteratively generates a series of "partial" programs (i.e., programs with missing code) over the library $\mathcal{L}$, starting with an "empty" program and ending with a complete program. A type inference procedure is used to rule out any partial program that is not type-safe. A cost heuristic is used to generate programs in an order of structural simplicity. Concretely, shorter programs are evaluated first.
- The second method is an *evolutionary algorithm* inspired by work on functional genetic programming [9]. Here, we use selection, crossover, and mutation operators to evolve a population of programs over $\mathcal{L}$. Types play a key role: all programs in the population are ensured to be type-safe, and mutation and crossover only replace a subterm in a program with terms of the same type.

In both cases, the use of types vastly reduces the amount of search that is needed, as the number of type-safe programs of a given size is a small fraction of the number of programs of that size. See Section 4 for an experimental confirmation.

**Lifelong Learning.** A *lifelong learning* setting is a sequence of related tasks $\mathcal{D}_1, \mathcal{D}_2, \ldots$, where each task $\mathcal{D}_i$ has its own training and validation set. Here, the learner is called repeatedly, once for each task $\mathcal{D}_i$ using a neural library $\mathcal{L}_i$, returning a best-performing program $e_i^*$ with parameters $v_i^*$.

We implement transfer learning simply by adding new modules to the neural library after each call to the learner. We add all neural functions from $e_i^*$ back into the library, freezing their parameters. More formally, let $\oplus_{i1} \ldots \oplus_{iK}$ be the neural library functions which are called anywhere in $e_i^*$. Each library function $\oplus_{ik}$ has parameters $w_{ik}$, set to the value $v_{ik}^*$ by TUNE. The library for the next task is then $\mathcal{L}_{i+1} = \mathcal{L}_i \cup \{\oplus_{ik}[w_{ik} \mapsto v_{ik}^*]\}$. This process ensures that the parameter vectors of $\oplus_{ik}$ are frozen and can no longer be updated by subsequent tasks. Thus, we prevent catastrophic forgetting by design. Importantly, it is always possible for the synthesizer to introduce "fresh networks" whose parameters have not been pretrained. This is because the library always monotonically increases over time, so that an original neural library function with untrained parameters is still available.

This approach has the important implication that the set of neural library functions that the synthesizer uses is not fixed, but continually evolving. Because both trained and untrained versions of the library functions are available, this can be seen to permit *selective transfer*, meaning that depending on which version of the library function GENERATE chooses, the learner has the option of using or not using previously learned knowledge in a new task. This fact allows HOUDINI to avoid negative transfer.

## 4 Evaluation

Our evaluation studies four questions. First, we ask whether HOUDINI can learn nontrivial differentiable programs that combine perception and algorithmic reasoning. Second, we study if HOUDINI can transfer perceptual and algorithmic knowledge during lifelong learning. We study three forms of transfer: *low-level transfer* of perceptual concepts across domains, *high-level transfer* of algorithmic concepts, and *selective transfer* where the learning method decides on which known concepts to

**Individual tasks**

recognize_digit(d): Binary classification of whether image contains a digit $d \in \{0 \dots 9\}$

classify_digit: Classify a digit into digit categories $(0-9)$

recognize_toy(t): Binary classification of whether an image contains a toy $t \in \{0 \dots 4\}$

regress_speed: Return the speed value and a maximum distance constant from an image of a speed limit sign.

regress_mnist: Return the value and a maximum distance constant from a digit image from MNIST dataset.

count_digit(d): Given a list of images, count the number of images of digit $d$

count_toy(t): Given a list of images, count the number of images of toy $t$

sum_digits: Given a list of digit images, compute the sum of the digits.

shortest_path_street: Given a grid of images of speed limit signs, find the shortest distances to all other nodes

shortest_path_mnist: Given a grid of MNIST images, and a source node, find the shortest distances to all other nodes in the grid.

**Task Sequences**

*Counting*

**CS1:** Evaluate low-level transfer.
*Task 1*: recognize_digit(d1); *Task 2*: recognize_digit(d2); *Task 3*: count_digit(d1); *Task 4*: count_digit(d2)

**CS2:** Evaluate high-level transfer, and learning of perceptual tasks from higher-level supervision.
*Task 1*: recognize_digit(d1); *Task 2*: count_digit(d1); *Task 3*: count_digit(d2); *Task 4*: recognize_digit(d2)

**CS3:** Evaluate high-level transfer of counting across different image domains.
*Task 1*: recognize_digit(d); *Task 2*: count_digit(d); *Task 3*: count_toy(t); *Task 4*: recognize_toy(t)

*Summing*

**SS:** Demonstrate low-level transfer of a multi-class classifier as well as the advantage of functional methods like foldl in specific situations.
*Task 1*: classify_digit; *Task 2*: sum_digits

*Single-Source Shortest Path*

**GS1:** Learning of complex algorithms.
*Task 1*: regress_speed; *Task 2*: shortest_path_street

**GS2:** High-level transfer of complex algorithms.
*Task 1*: regress_mnist; *Task 2*: shortest_path_mnist; *Task 3*: shortest_path_street

*Long sequence* **LS**.
*Task 1*:count_digit(d1); *Task 2*: count_toy(t1); *Task 3*: recognize_toy(t2); *Task 4*: recognize_digit(d2); *Task 5*: count_toy(t3); *Task 6*: count_digit(d3); *Task 7*: count_toy(t4); *Task 8*: recognize_digit(d4); *Task 9*: count_digit(d5)

Figure 3: Tasks and task sequences.

reuse. Third, we study the value of our type-directed approach to synthesis. Fourth, we compare the performance of the top-down and evolutionary synthesis algorithms.

**Task Sequences.** Each lifelong learning setting is a sequence of individual learning tasks. The full list of tasks is shown in Figure 3. These tasks include object recognition tasks over three data sets: MNIST [21], NORB [22], and the GTSRB data set of images of traffic signs [37]. In addition, we have three algorithmic tasks: *counting* the number of instances of images of a certain class in a list of images; *summing* a list of images of digits; and the *shortest path* computation described in Section 2.

We combine these tasks into seven sequences. Three of these (CS1, SS, GS1) involve low-level transfer, in which earlier tasks are perceptual tasks like recognizing digits, while later tasks introduce higher-level algorithmic problems. Three other task sequences (CS2, CS3, GS2) involve higher-level transfer, in which earlier tasks introduce a high-level concept, while later tasks require a learner to re-use this concept on different perceptual inputs. For example, in CS2, once count_digit($d_1$) is learned for counting digits of class $d_1$, the synthesizer can learn to reuse this counting network on a new digit class $d_2$, even if the learning system has never seen $d_2$ before. The graph task sequence GS1 also demonstrates that the graph convolution combinator in HOUDINI allows learning of complex graph algorithms and GS2 tests if high-level transfer can be performed with this more complex task. Finally, we include a task sequence LS that is designed to evaluate our method on a task sequence that is both longer and that lacks a favourable curriculum. The sequence LS was initially randomly generated, and then slightly amended in order to evaluate all lifelong learning concepts discussed.

**Experimental setup.** We allow three kinds of neural library modules: multi-layer perceptrons (MLPs), convolutional neural networks (CNNs) and recurrent neural networks (RNNs). We use two symbolic synthesis strategies: top-down refinement and evolutionary. We use three types of baselines: (1) *standalone networks*, which do not do transfer learning, but simply train a new network (an RNN) for each task in the sequence, starting from random weights; (2) a traditional neural approach to *low-level transfer* (LLT) that transfers all weights learned in the previous task, except for the output layer that is kept task-specific; and (3) a version of the *progressive neural networks* (PNNs) [32]

| Task | Top 3 programs | RMSE |
|---|---|---|
| Task 1: regress_mnist | 1. nn_gs2_1 ∘ nn_gs2_2 | 1.47 |
| Task 2: shortest_path_mnist | 1. ($\mathbf{conv\_g}^{10}$ (nn_gs2_3)) ∘ ($\mathbf{map\_g}$ (lib.nn_gs2_1 ∘ lib.nn_gs2_2)) | 1.57 |
| | 2. ($\mathbf{conv\_g}^{9}$ (nn_gs2_4)) ∘ ($\mathbf{map\_g}$ (lib.nn_gs2_1 ∘ lib.nn_gs2_2)) | 1.72 |
| | 3. ($\mathbf{conv\_g}^{9}$ (nn_gs2_5)) ∘ ($\mathbf{map\_g}$ (nn_gs2_6 ∘ nn_gs2_7)) | 4.99 |
| Task 3: shortest_path_street | 1. ($\mathbf{conv\_g}^{10}$(lib.nn_gs2_3)) ∘ ($\mathbf{map\_g}$ (nn_gs2_8 ∘ nn_gs2_9)) | 3.48 |
| | 2. ($\mathbf{conv\_g}^{9}$(lib.nn_gs2_3)) ∘ ($\mathbf{map\_g}$ (nn_gs2_10 ∘ nn_gs2_11)) | 3.84 |
| | 3. ($\mathbf{conv\_g}^{10}$(lib.nn_gs2_3)) ∘ ($\mathbf{map\_g}$ (lib.nn_gs2_1 ∘ lib.nn_gs2_2)) | 6.91 |

Figure 4: Top 3 synthesized programs on Graph Sequence 2 (GS2). $\mathbf{map\_g}$ denotes a graph map (of the appropriate type); $\mathbf{conv\_g}^{i}$ denotes $i$ repeated applications of a graph convolution combinator.

approach, which retains a pool of pretrained models during training and learns lateral connections among these models. Experiments were performed using a single-threaded implementation on a Linux system, with 8-core Intel E5-2620 v4 2.10GHz CPUs and TITAN X (Pascal) GPUs.

The architecture chosen for the standalone and LLT baselines composes an MLP, an RNN, and a CNN, and matches the structure of a high-performing program returned by HOUDINI, to enable an apples-to-apples comparison. In PNNs, every task in a sequence is associated with a network with the above architecture; lateral connections between these networks are learned. Each sequence involving digit classes $d$ and toy classes $t$ was instantiated five times for random values of $d$ and $t$, and the results shown are averaged over these instantiations. In the graph sequences, we ran the same sequences with different random seeds, and shared the regressors learned for the first tasks across the competing methods for a more reliable comparison. We do not compare against PNNs in this case, as it is nontrivial to extend them to work with graphs. We evaluate the competing approaches on 2%, 10%, 20%, 50% and 100% of the training data for all but the graph sequences, where we evaluate only on 100%. For classification tasks, we report error, while for the regression tasks — counting, summing, regress_speed and shortest_path — we report root mean-squared error (RMSE).

**Results: Synthesized programs.** HOUDINI successfully synthesizes programs for each of the tasks in Figure 3 within at most 22 minutes. We list in Figure 4 the top 3 programs for each task in the graph sequence GS2, and the corresponding RMSEs. Here, function names with prefix "nn_" denote fresh neural modules trained during the corresponding tasks. Terms with prefix "lib." denote pretrained neural modules selected from the library. The synthesis times for Task 1, Task 2, and Task 3 are 0.35s, 1061s, and 1337s, respectively.

As an illustration, consider the top program for Task 3: ($\mathbf{conv\_g}^{10}$ lib.nn_gs2_3) ∘ ($\mathbf{map\_g}$ (nn_gs2_8 ∘ nn_gs2_9)). Here, $\mathbf{map\_g}$ takes as argument a function for processing the images of speed limits. Applied to the input graph, the map returns a graph $G$ in which each node contains a number associated with its corresponding image and information about the least cost of travel to the node. The kernel for the graph convolution combinator $\mathbf{conv\_g}$ is a function lib.nn_gs2_3, originally learned in Task 2, that implements the *relaxation* operation used in shortest-path algorithms. The convolution is applied repeatedly, just like in the Bellman-Ford shortest path algorithm.

In the SS sequence, the top program for Task 2 is: ($\mathbf{fold\_l}$ nn_ss_3 zeros(1)) ∘ $\mathbf{map\_l}$(nn_ss_4 ∘ lib.nn_ss_2). Here, $\mathbf{fold\_l}$ denotes the fold operator applied to lists, and zeros(dim) is a function that returns a zero tensor of appropriate dimension. The program uses a map to apply a previously learned CNN feature extractor (lib.nn_ss_2) and a learned transformation of said features into a 2D hidden state, to all images in the input list. It then uses fold with another function (nn_ss_3) to give the final sum. Our results, presented in the supplementary material, show that this program greatly outperforms the baselines, even in the setting where all of the training data is available. We believe that this is because the synthesizer has selected a program with fewer parameters than the baseline RNN. In the results for the counting sequences (CS) and the long sequence (LS), the number of evaluated programs is restricted to 20, therefore $\mathbf{fold\_l}$ is not used within the synthesized programs. This allows us to evaluate the advantage of HOUDINI brought by its transfer capabilities, rather than its rich language.

**Results: Transfer.** First we evaluate the performance of the methods on the counting sequences (Figure 5). For space, we omit early tasks where, by design, there is no opportunity for transfer; for these results, see the Appendix. In all cases where there is an opportunity to transfer from previous tasks, we see that HOUDINI has much lower error than any of the other transfer learning methods. The actual programs generated by HOUDINI are listed in the Appendix.

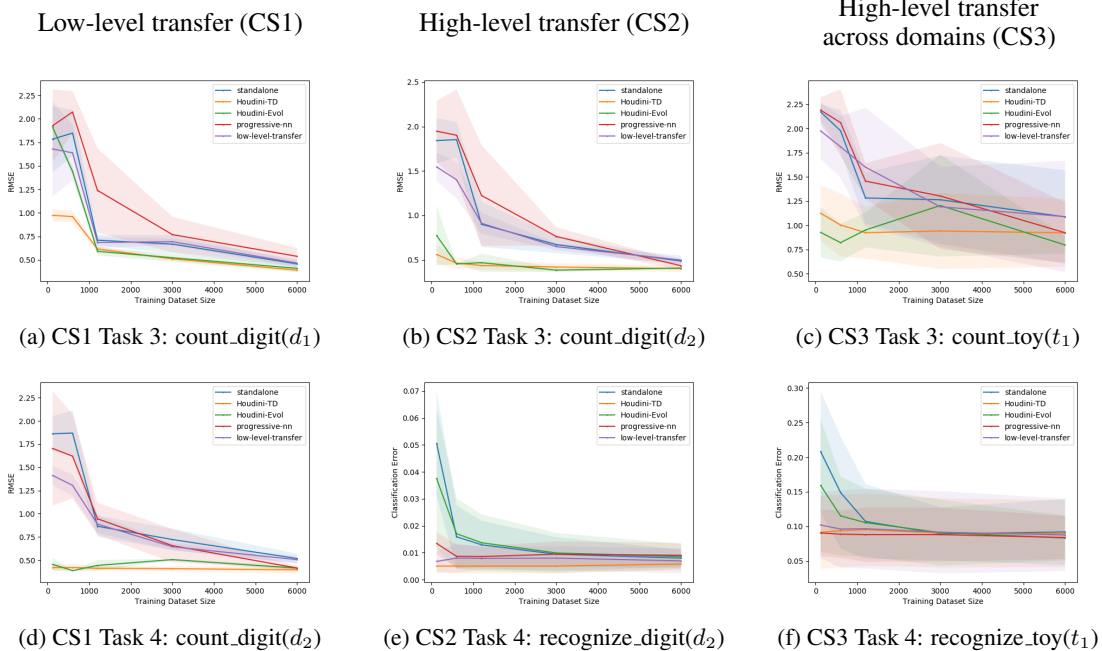

| Low-level transfer (CS1) | High-level transfer (CS2) | High-level transfer across domains (CS3) |
|---|---|---|
| (a) CS1 Task 3: count_digit($d_1$) | (b) CS2 Task 3: count_digit($d_2$) | (c) CS3 Task 3: count_toy($t_1$) |
| (d) CS1 Task 4: count_digit($d_2$) | (e) CS2 Task 4: recognize_digit($d_2$) | (f) CS3 Task 4: recognize_toy($t_1$) |

Figure 5: Lifelong "learning to count" (Sequences CS1 – CS3), demonstrating both low-level transfer of perceptual concepts and high-level transfer of a counting network. HOUDINI-TD and HOUDINI-EVOL are HOUDINI with the top-down and evolutionary synthesizers, respectively.

Task sequence CS1 evaluates the method's ability to selectively perform low-level transfer of a perceptual concept across higher level tasks. The first task that provides a transfer opportunity is CS1 task 3 (Figure 5a). There are two potential lower-level tasks that the methods could transfer from: recognize_digit($d_1$) and recognize_digit($d_2$). HOUDINI learns programs composed of neural modules nn_cs1_1, nn_cs1_2, nn_cs1_3, and nn_cs1_4 for these two tasks. During training for the count_digit($d1$) task, all the previously learned neural modules are available in the library. The learner, however, picks the correct module (nn_cs1_2) for reuse, learning the program "nn_cs1_7 ∘ (**map_l** (nn_cs1_8 ∘ lib.nn_cs1_2))" where nn_cs1_7 and nn_cs1_8 are fresh neural modules, and **map_l** stands for a list map combinator of appropriate type. The low-level transfer baseline cannot select which of the previous tasks to re-use, and so suffers worse performance.

Task sequence CS2 provides an opportunity to transfer the higher-level concepts of counting, across different digit classification tasks. Here CS2 task 3 (Figure 5b) is the task that provides the first opportunity for transfer. We see that HOUDINI is able to learn much faster on this task because it is able to reuse a network which has learned from the previous counting task. Task sequence CS3 examines whether the methods can demonstrate high-level transfer when the input image domains are very different, from the MNIST domain to the NORB domain of toy images. We see in Figure 5c that the higher-level network still successfully transfers across tasks, learning an effective network for counting the number of toys of type $t_1$, even though the network has not previously seen any toy images at all. What is more, it can be seen that because of the high-level transfer, HOUDINI has learned a modular solution to this problem. From the subsequent performance on a standalone toy classification task (Figure 5f), we see that CS3 task 3 has already caused the network to induce a re-usable classifier on toys. Overall, it can be seen that HOUDINI outperforms all the baselines even under the limited data setting, confirming the successful selective transfer of both low-level and high-level perceptual information. Similar results can be seen on the summing task (see supplementary material). Moreover, on the longer task sequence LS, we also find that HOUDINI performs significantly better on the tasks in the sequence where there is an opportunity for transfer, and performs comparably the baselines on the other tasks (see supplementary material). Furthermore, on the summing sequence, our results also show low level transfer.

Finally, for the graph-based tasks (Table 2), we see that the graph convolutional program learned by HOUDINI on the graph tasks has significantly less error than a simple sequence model, a standalone

| Task | Number of programs | | |
|---|---|---|---|
| | size = 4 | size = 5 | size = 6 |
| **No types** | | | |
| Task 1 | 8182 | 110372 | 1318972 |
| Task 2 | 12333 | 179049 | 2278113 |
| Task 3 | 17834 | 278318 | 3727358 |
| Task 4 | 24182 | 422619 | 6474938 |
| **+ Types** | | | |
| Task 1 | 2 | 20 | 44 |
| Task 2 | 5 | 37 | 67 |
| Task 3 | 9 | 47 | 158 |
| Task 4 | 9 | 51 | 175 |

Table 1: Effect of the type system on the number of programs considered in the symbolic search for task sequence CS1.

| | Task 1 | Task 2 |
|---|---|---|
| RNN w llt | 0.75 | 5.58 |
| standalone | 0.75 | 4.96 |
| HOUDINI | 0.75 | 1.77 |
| HOUDINI EA | 0.75 | 8.32 |
| low-level-transfer | 0.75 | 1.98 |

(a) Low-level transfer (llt) (task sequence GS1).

| | Task 1 | Task 2 | Task 3 |
|---|---|---|---|
| RNN w llt | 1.44 | 5.00 | 6.05 |
| standalone | 1.44 | 6.49 | 7. |
| HOUDINI | 1.44 | 1.50 | 3.31 |
| HOUDINI EA | 1.44 | 6.67 | 7.88 |
| low-level-transfer | 1.44 | 1.76 | 2.08 |

(b) High-level transfer (task sequence GS2).

Table 2: Lifelong learning on graphs. Col 1: RMSE on speed/distance from image. Cols 2, 3: RMSE on shortest path (mnist, street).

baseline and the evolutionary-algorithm-based version of HOUDINI. As explained earlier, in the shortest_path_street task in the graph sequence GS2, HOUDINI learns a program that uses newly learned regress functions for the street signs, along with a "relaxation" function already learned from the earlier task shortest_path_mnist. In Table 2, we see this program performs well, suggesting that a domain-general relaxation operation is being learned. Our approach also outperforms the low-level-transfer baseline, except on the shortest_path_street task in GS2. We are unable to compare directly to NTPT because no public implementation is available. However, our graph task is a more difficult version of a task from [14], who report on their shortest-path task "2% of random restarts successfully converge to a program that generalizes" (see their supplementary material).

**Results: Typed vs. untyped synthesis.** To assess the impact of our type system, we count the programs that GENERATE produces with and without a type system (we pick the top-down implementation for this test, but the results also apply to the evolutionary synthesizer). Let the *size* of a program be the number of occurrences of library functions and combinators in the program. Table 1 shows the number of programs of different sizes generated for the tasks in the sequence CS1. Since the typical program size in our sequences is less than 6, we vary the target program size from 4 to 6. When the type system is disabled, the only constraint that GENERATE has while composing programs is the arity of the library functions. We note that this constraint fails to bring down the number of candidate programs to a manageable size. With the type system, however, GENERATE produces far fewer candidate programs. For reference, neural architecture search often considers thousands of potential architectures for a single task [24].

**Results: Top-Down vs. Evolutionary Synthesis.** Overall, the top-down implementation of GENERATE outperformed the evolutionary implementation. In some tasks, the two strategies performed similarly. However, the evolutionary strategy has high variance; indeed, in many runs of the task sequences, it times out without finding a solution. The timed out runs are not included in the plots.

# 5 Conclusion

We have presented HOUDINI, the first neurosymbolic approach to the synthesis of differentiable functional programs. Deep networks can be naturally specified as differentiable programs, and functional programs can compactly represent popular deep architectures [26]. Therefore, symbolic search through a space of differentiable functional programs is particularly appealing, because it can at the same time select both which pretrained neural library functions should be reused, and also what deep architecture should be used to combine them. On several lifelong learning tasks that combine perceptual and algorithmic reasoning, we showed that HOUDINI can accelerate learning by transferring high-level concepts.

**Acknowledgements.** This work was partially supported by DARPA MUSE award #FA8750-14-2-0270 and NSF award #CCF-1704883.

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
