[Supplementary Material · HOUDINI_lifelong_learning_supplementary.pdf]

# Supplementary material for HOUDINI: Lifelong Learning as Program Synthesis

**Lazar Valkov**
University of Edinburgh
L.Valkov@sms.ed.ac.uk

**Dipak Chaudhari**
Rice University
dipakc@rice.edu

**Akash Srivastava**
University of Edinburgh
Akash.Srivastava@ed.ac.uk

**Charles Sutton**
University of Edinburgh,
The Alan Turing Institute, and Google Brain
charlessutton@google.com

**Swarat Chaudhuri**
Rice University
swarat@rice.edu

## A   Assigning Types to HOUDINI Programs

A program $e$ in HOUDINI is assigned a type using the following rules:

- $e = e' \circ e''$ is assigned a type iff $e'$ has type $\tau\tau'$ and $e''$ has type $\tau'\tau''$. In this case, $e$ has type $\tau\tau''$.
- $e =_{\alpha\tau} e'$ is assigned a type iff $e'$ has the type $\tau\tau'$. In this case, the type of $e$ is $\alpha\tau\alpha\tau'$.
- $e =_{\alpha\tau} e' \, z$ is assigned a type iff $e'$ has the type $\tau'(\tau\tau')$ and $z$ has the type $\tau'$. In this case, $e$ has type $\alpha\tau\tau'$.
- $e =_{\alpha\tau} e'$ is assigned a type iff $e'$ has the type $\texttt{list}\tau\tau'$. In this case, $e$ has type $\alpha\tau\alpha\tau'$.

If it is not possible to assign a type to the program $e$, then it is considered *type-inconsistent* and excluded from the scope of synthesis.

## B   Symbolic Program Synthesis

In this appendix we provide implementation details of our synthesis algorithms.

### B.1   Synthesis Using Top-down Iterative Refinement

Now we give more details on the implementation of based on iterative refinement. To explain this algorithm, we need to define a notion of a *partial program*. The grammar for partial programs $e$ is obtained by augmenting the HOUDINI grammar (Figure **??**) with an additional rule: $e ::= \tau$. The form $\tau$ represents a *hole*, standing for missing code. A program with holes has no operational meaning; however, we do have a type system for such programs. This type system follows the rules in Appendix A, but in addition, axiomatically assumes any subterm $\tau$ to be of type $\tau$. A partial program that cannot be assigned a type is automatically excluded from the scope of synthesis.

Now, the initial input to the algorithm is the type $\tau$ of the function we want to learn. The procedure proceeds iteratively, maintaining a priority queue $Q$ of *synthesis subtasks* of the form $(e, f)$, where $e$ is a type-safe partial or complete program of type $\tau$, and $f$ is either a hole of type $\tau'$ in $e$, or a special symbol $\perp$ indicating that $e$ is complete (i.e., free of holes). The interpretation of such a task is to find a replacement $e'$ of type $\tau'$ for the hole $f$ such that the program $e''$ obtained by substituting $f$ by $e'$ is complete. (Because $e$ is type-safe by construction, $e''$ is of type $\tau$.) The queue is sorted according to a heuristic cost function that prioritizes simpler programs.

Initially, $Q$ has a single element $(e, f)$, where $e$ is an "empty" program of form $\tau$, and $f$ is a reference to the hole in $e$. The procedure iteratively processes subtasks in the queue $Q$, selecting a task $(e, f)$

in the beginning of each iteration. If the program $e$ is complete, it is sent to the neural module for parameter learning. Otherwise, the algorithm expands the program $e$ by proposing a partial program that fills the hole $f$. To do this, the algorithm selects a production rule for partial programs from the grammar for partial programs. Suppose the right hand side of this rule is $\alpha$. The algorithm constructs an expression $e'$ from $\alpha$ by replacing each nonterminal in $\alpha$ by a hole with the same type as the nonterminal. If $e'$ is not of the same type as $f$, it is automatically rejected. Otherwise, the algorithm constructs the program $e'' = e[f \mapsto e']$. For each hole $f'$ in $e''$, the algorithm adds to $Q$ a new task $(e'', f')$. If $e''$ has no hole, it adds to $Q$ a task $(e'', \perp)$.

## B.2 Evolutionary Synthesis

The evolutionary synthesis algorithm is an iterative procedure that maintains a population of programs. The population is initialized with a set of randomly generated type-safe parameterized programs. Each iteration of the algorithm performs the following steps.

1. Each program in the population is sent to the neural module , which computes a *fitness score* (the loss under optimal parameters) for the program.

2. We perform *random proportional selection*, in which a subset of the (parameterized) programs are retained, while the other programs are filtered out. Programs with higher fitness are more likely to remain in the population.

3. We perform a series of *crossover* operations, each of which draws a random pair of programs from the population and swaps a pair of randomly drawn subterms of the same type in these programs.

4. We perform a series of *mutation* operations, each of which randomly chooses a program and replaces a random subterm the program with a new subterm of the same type.

Because the crossover and mutation operations only replace terms with other terms of the same type, the programs in the population are always guaranteed to be type-consistent. This fact is key to the performance of the algorithm.

## C Details of Experimental Setup

The initial library models, which have trainable weights, have the following architecture. MLP modules have one hidden layer of size 1024, followed by batch normalization and dropout, followed by an output layer. CNNs have two convolutional layers with 32 and 64 output channels respectively, each with a 5x5 kernel, stride 1 and 0 padding, and each followed by max pooling, followed by spatial dropout. RNN modules are long short-term memory (LSTM) networks with a hidden dimension of 100, followed by an output layer, which transforms the last hidden state. For a given task, we use the input and output types of the new function to decide between MLP, CNN, or RNN, and also deduce the output activation function.

The standalone baseline for counting uses an architecture of the form $\lambda x.\text{RNN}(\,(\text{MLP} \circ \text{CNN}(x)))$, which is intuitively appropriate for the task, and also matches the shape of some programs commonly returned by HOUDINI. As for the shortest path sequences, the first task for GS1 and GS2 is regression, which we train using a network with architecture $\text{MLP} \circ \text{CNN}$, in which the last layer is linear. In the RNN baseline for the other tasks in the graph sequences, we map a learned $\text{MLP} \circ \text{CNN}$ regression module to each image in the grid. Afterwards, we linearize the grid row-wise, converting it into a list, and then we process it using an LSTM (RNN) with hidden state of size 100. The number was chosen so that both our implementation and the baseline have almost the same number of parameters.

For multi-class classification (Sequence SS - Task 1) and regression (GS1 - Task1, GS2 - Task 1), we used all training images available. For the rest of the tasks in GS1, GS2, GS3 and SS, we use 12000 data points for training, with 2100 for testing. The list lengths for training are [2, 3, 4, 5], and [6, 7, 8] for testing in order to evaluate the generalization to longer sequences. We train for 20 epochs on all list-related tasks and for 1000 epochs for the regression tasks. The training datasets for the graph shortest path tasks (GS1 - Task 2; GS2 - Task2, GS2 - Task3) consists of 70,000 3x3 grids and 1,000,000 4x4 grids, while the testing datasets consists of 10,000 5x5 grids. The number of epochs for these tasks is 5. In GS2 - Task3, the *low-level transfer* baseline reuses the regression function learned in GS2 - Task1, thus, the image dimensions from MNIST and the colored GTSRB need to

match. Therefore, we expanded the MNIST digit images, used for the graph sequences GS1 and GS2, to 28x28x3 dimensionality and resized the images from GTSRB from 32x32x3 to 28x28x3.

For all experiments, we use early stopping, reporting the the test error at the epoch where the validation error was minimized.

## D  Programs Discovered in Experiments

Tables 1-22 list the top 3 programs and the corresponding classification errors/RMSEs, on a test dataset, for most of our task sequences. The programs are ordered by their performance on a validaiton dataset. Finally, the presented programs are the ones evaluated for all (100%) of the training dataset. Here we use the syntax to denote function composition. Program terms with prefix "nn_" denote neural modules trained during the corresponding tasks whereas terms with prefix "lib." denote already trained neural modules in the library. For example, in Counting Sequence 1 (Table 1), "nn_cs1_1" is the neural module trained during Task 1 (recognize_digit($d_1$)). After completion of this task, the neural module is added to the library and is available for use during the subsequent tasks. For example, the top performing program for Task 3 (count_digit($d_1$)) uses the neural module "lib.nn_cs1_1" from the library (and a freshly trained neural module "nn_cs1_5") to construct a program for the counting task.

## E  Summing Experiment

In this section we present the result from task sequence SS in Figure 3 of the main paper. This sequence was designed to demonstrate low-level transfer of a multi-class classifier as well as the advantage of functional methods like foldl in specific situations. The first task of the sequence is a simple MNIST classifier, on which all competing methods do equally well. The second task is a regression task, to learn to sum all of the digits in the sequence. The standalone method, low level transfer one and the progressive neural networks all perform equally poorly (note that their lines are overplotted in the Figure), but the synthesized program from HOUDINI is able to learn this function easily because it is able to use a foldl operation. We also add a new baseline "standalone_with_fold", which reuses the program found by HOUDINI, but trains the parameter from a random initialization.

(a) Task 2: Sum digits

Figure 1: Lifelong learning for "learning to sum" (Sequence SS).

## F  Full Experimental Results on Counting Tasks

In the paper, we present results for the counting sequences on for the later tasks, in which transfer learning is possible. For completeness, in this section we present results on all of the tasks in the sequences. See Figures 2–4. We note that for the early tasks in each task sequence (e.g. CS1 tasks 1 and 2), there is little relevant information that can be transferred from early tasks, so as expected all methods perform similarly; e.g., the output of HOUDINI is a single library function.

Table 1: Counting Sequence 1(CS1). "CE" denotes classification error and "RMSE" denotes root mean square error.

| Task | Top 3 programs | Error |
|---|---|---|
| Task 1: recognize_digit($d_1$) | 1. (nn_cs1_1, nn_cs1_2) | 1% CE |
| Task 2: recognize_digit($d_2$) | 1. (nn_cs1_3, nn_cs1_4) | 1% CE |
| | 2. (nn_cs1_5, lib.nn_cs1_2) | 1% CE |
| | 3. (lib.nn_cs1_1, nn_cs1_6) | 1% CE |
| Task 3: count_digit($d_1$) | 1. (nn_cs1_7, ((nn_cs1_8, lib.nn_cs1_2))) | 0.38 RMSE |
| | 2. ((nn_cs1_9, (nn_cs1_10)), (nn_cs1_11)) | 0.38 RMSE |
| | 3. (nn_cs1_12, ((nn_cs1_13), (lib.nn_cs1_2))) | 0.40 RMSE |
| Task 4: count_digit($d_2$) | 1. (nn_cs1_14, ((lib.nn_cs1_1), (nn_cs1_15))) | 0.32 RMSE |
| | 2. (lib.nn_cs1_7, ((nn_cs1_16, lib.nn_cs1_4))) | 0.37 RMSE |
| | 3. (lib.nn_cs1_7, ((nn_cs1_17), (lib.nn_cs1_4))) | 0.37 RMSE |

Table 2: Counting Sequence 2(CS2)

| Task | Top 3 programs | Error |
|---|---|---|
| Task 1: recognize_digit($d_1$) | 1. (nn_cs2_1, nn_cs2_2) | 1% CE |
| Task 2: count_digit($d_1$) | 1. (nn_cs2_3, ((nn_cs2_4), (nn_cs2_5))) | 0.35 RMSE |
| | 2. (nn_cs2_6, ((nn_cs2_7, nn_cs2_8))) | 0.40 RMSE |
| | 3. ((nn_cs2_9, (nn_cs2_10)), (nn_cs2_11)) | 0.41 RMSE |
| Task 3: count_digit($d_2$) | 1. (lib.nn_cs2_3, ((nn_cs2_12), (lib.nn_cs2_2))) | 0.34 RMSE |
| | 2. (lib.nn_cs2_3, ((nn_cs2_13, lib.nn_cs2_2))) | 0.33 RMSE |
| | 3. (lib.nn_cs2_3, ((nn_cs2_14), (nn_cs2_15))) | 0.33 RMSE |
| Task 4: recognize_digit($d_2$) | 1. (nn_cs2_16, nn_cs2_17) | 1% CE |
| | 2. (nn_cs2_18, lib.nn_cs2_2) | 1% CE |
| | 3. (lib.nn_cs2_12, lib.nn_cs2_2) | 1% CE |

Table 3: Counting Sequence 3(CS3)

| Task | Top 3 Programs | Error |
|---|---|---|
| Task 1: recognize_digit($d$) | 1. (nn_cs3_1, nn_cs3_2) | 1% CE |
| Task 2: count_digit($d$) | 1. (nn_cs3_3, ((nn_cs3_4, nn_cs3_5))) | 0.40 RMSE |
| | 2. (nn_cs3_6, ((nn_cs3_7, lib.nn_cs3_2))) | 0.40 RMSE |
| | 3. (nn_cs3_8, ((nn_cs3_9), (lib.nn_cs3_2))) | 0.41 RMSE |
| Task 3: count_toy($t$) | 1. (lib.nn_cs3_3, ((nn_cs3_10), (nn_cs3_11))) | -0.73 |
| | 2. (lib.nn_cs3_3, ((nn_cs3_12, nn_cs3_13))) | 0.67 RMSE |
| | 3. (lib.nn_cs3_3, ((lib.nn_cs3_1), (nn_cs3_14))) | 0.96 RMSE |
| Task 4: recognize_toy($t$) | 1. (nn_cs3_15, lib.nn_cs3_11) | 7% CE |
| | 2. (nn_cs3_16, nn_cs3_17) | 5% CE |
| | 3. (lib.nn_cs3_10, lib.nn_cs3_11) | 6% CE |

Table 4: Summing Sequence(SS)

| Task | Top 3 programs | Error |
|---|---|---|
| Task 1: classify_digit | 1. (nn_ss_1, nn_ss_2) | 1% CE |
| Task 2: sum_digits | 1. (( nn_ss_3 zeros(1)), ((nn_ss_4, lib.nn_ss_2))) | 2.15 RMSE |
| | 2. (( nn_ss_5 zeros(1)), ((nn_ss_6, nn_ss_7))) | 2.58 RMSE |
| | 3. (( nn_ss_8 zeros(1)), ((lib.nn_ss_1, lib.nn_ss_2))) | 4.30 RMSE |

Table 5: Graph Sequence 1(GS1)

| Task | Top 3 Programs | Error |
|---|---|---|
| Task 1: regress_speed | 1. (nn_gs1_1, nn_gs1_2) | 0.64 RMSE |
| Task 2: shortest_path_street | 1. ($^{10}$(nn_gs1_3), ((lib.nn_gs1_1, lib.nn_gs1_2))) | 1.88 RMSE |
| | 2. ($^{9}$(nn_gs1_4), ((lib.nn_gs1_1, lib.nn_gs1_2))) | 2.02 RMSE |
| | 3. ($^{10}$(nn_gs1_5), ((nn_gs1_6, nn_gs1_7))) | 6.76 RMSE |

Table 6: Graph Sequence 2(GS2)

| Task | Top 3 Programs | Error |
|---|---|---|
| Task 1: regress_mnist | 1. (nn_gs2_1, nn_gs2_2) | 1.47 RMSE |
| Task 2: shortest_path_mnist | 1. ($^{10}$(nn_gs2_3), ((lib.nn_gs2_1, lib.nn_gs2_2))) | 1.57 RMSE |
| | 2. ($^{9}$(nn_gs2_4), ((lib.nn_gs2_1, lib.nn_gs2_2))) | 1.73 RMSE |
| | 3. ($^{9}$(nn_gs2_5), ((nn_gs2_6, nn_gs2_7))) | 4.99 RMSE |
| Task 3: shortest_path_street | 1. ($^{10}$(lib.nn_gs2_3), ((nn_gs2_8, nn_gs2_9))) | 3.48 RMSE |
| | 2. ($^{9}$(lib.nn_gs2_3), ((nn_gs2_10, nn_gs2_11))) | 3.84 RMSE |
| | 3. ($^{10}$(lib.nn_gs2_3), ((lib.nn_gs2_1, lib.nn_gs2_2))) | 6.92 RMSE |

Table 7: Long Sequence 1(LS1).

| Task | Top 3 Programs | Error |
|---|---|---|
| Task 1: count_digit(7) | 1. ((nn_ls1_1, (nn_ls1_2)), (nn_ls1_3)) | 0.46 RMSE |
| | 2. (nn_ls1_4, ((nn_ls1_5, nn_ls1_6))) | 0.49 RMSE |
| | 3. (nn_ls1_7, ((nn_ls1_8), (nn_ls1_9))) | 0.51 RMSE |
| Task 2: count_digit(4) | 1. (lib.nn_ls1_1, ((nn_ls1_10), (nn_ls1_11))) | 1.50 RMSE |
| | 2. (lib.nn_ls1_1, ((nn_ls1_12, nn_ls1_13))) | 1.61 RMSE |
| | 3. ((lib.nn_ls1_1, (nn_ls1_14)), (nn_ls1_15)) | 1.64 RMSE |
| Task 3: recognize_toy(0) | 1. (nn_ls1_16, lib.nn_ls1_11) | 9.81% CE |
| | 2. (nn_ls1_17, nn_ls1_18) | 8.86% CE |
| | 3. (nn_ls1_19, lib.nn_ls1_3) | 12.86% CE |
| Task 4: recognize_digit(9) | 1. (nn_ls1_20, nn_ls1_21) | 1.38% CE |
| | 2. (nn_ls1_22, lib.nn_ls1_3) | 2.14% CE |
| | 3. (lib.nn_ls1_2, nn_ls1_23) | 1.95% CE |
| Task 5: count_digit(2) | 1. (nn_ls1_24, ((nn_ls1_25), (nn_ls1_26))) | 1.08 RMSE |
| | 2. (lib.nn_ls1_1, ((nn_ls1_27, nn_ls1_28))) | 1.02 RMSE |
| | 3. (lib.nn_ls1_1, ((lib.nn_ls1_16, nn_ls1_29))) | 0.95 RMSE |
| Task 6: count_digit(9) | 1. (lib.nn_ls1_1, ((nn_ls1_30, nn_ls1_31))) | 0.49 RMSE |
| | 2. (lib.nn_ls1_1, ((nn_ls1_32, lib.nn_ls1_21))) | 0.49 RMSE |
| | 3. (lib.nn_ls1_1, ((nn_ls1_33, lib.nn_ls1_3))) | 0.49 RMSE |
| Task 7: count_digit(0) | 1. (nn_ls1_34, ((lib.nn_ls1_16, lib.nn_ls1_11))) | 0.94 RMSE |
| | 2. (lib.nn_ls1_1, ((nn_ls1_35, nn_ls1_36))) | 0.81 RMSE |
| | 3. (nn_ls1_37, ((nn_ls1_38, nn_ls1_39))) | 0.85 RMSE |
| Task 8: recognize_digit(7) | 1. (lib.nn_ls1_2, lib.nn_ls1_3) | 0.86% CE |
| | 2. (nn_ls1_40, lib.nn_ls1_3) | 1.19% CE |
| | 3. (nn_ls1_41, lib.nn_ls1_21) | 1.05% CE |
| Task 9: count_digit(2) | 1. (nn_ls1_42, ((nn_ls1_43, lib.nn_ls1_26))) | 0.43 RMSE |
| | 2. (lib.nn_ls1_1, ((nn_ls1_44, nn_ls1_45))) | 0.45 RMSE |
| | 3. (lib.nn_ls1_1, ((nn_ls1_46, lib.nn_ls1_3))) | 0.45 RMSE |

Table 8: Long Sequence 2(LS2).

| Task | Top 3 Programs | Error |
|---|---|---|
| Task 1: count_digit(1) | 1. (nn_ls2_1, ((nn_ls2_2, nn_ls2_3))) | 0.43 RMSE |
| | 2. (nn_ls2_4, ((nn_ls2_5), (nn_ls2_6))) | 0.45 RMSE |
| | 3. ((nn_ls2_7, (nn_ls2_8)), (nn_ls2_9)) | 0.48 RMSE |
| Task 2: count_digit(0) | 1. (nn_ls2_10, ((nn_ls2_11), (nn_ls2_12))) | 0.96 RMSE |
| | 2. (lib.nn_ls2_1, ((nn_ls2_13, nn_ls2_14))) | 0.84 RMSE |
| | 3. ((lib.nn_ls2_1, (nn_ls2_15)), (nn_ls2_16)) | 0.92 RMSE |
| Task 3: recognize_toy(1) | 1. (nn_ls2_17, nn_ls2_18) | 5.05% CE |
| | 2. (nn_ls2_19, lib.nn_ls2_12) | 4.00% CE |
| | 3. (lib.nn_ls2_2, nn_ls2_20) | 10.52% CE |
| Task 4: recognize_digit(5) | 1. (nn_ls2_21, nn_ls2_22) | 0.76% CE |
| | 2. (nn_ls2_23, lib.nn_ls2_3) | 0.86% CE |
| | 3. (lib.nn_ls2_17, nn_ls2_24) | 0.81% CE |
| Task 5: count_digit(4) | 1. (lib.nn_ls2_1, ((nn_ls2_25, nn_ls2_26))) | 1.68 RMSE |
| | 2. (lib.nn_ls2_1, ((nn_ls2_27, lib.nn_ls2_18))) | 1.51 RMSE |
| | 3. (lib.nn_ls2_1, ((nn_ls2_28, lib.nn_ls2_12))) | 1.46 RMSE |
| Task 6: count_digit(5) | 1. (nn_ls2_29, ((nn_ls2_30, nn_ls2_31))) | 0.43 RMSE |
| | 2. (nn_ls2_32, ((lib.nn_ls2_21, lib.nn_ls2_22))) | 0.43 RMSE |
| | 3. (lib.nn_ls2_1, ((nn_ls2_33, lib.nn_ls2_22))) | 0.45 RMSE |
| Task 7: count_digit(1) | 1. (nn_ls2_34, ((lib.nn_ls2_25, nn_ls2_35))) | 0.64 RMSE |
| | 2. (nn_ls2_36, ((nn_ls2_37, nn_ls2_38))) | 0.74 RMSE |
| | 3. (nn_ls2_39, ((nn_ls2_40, lib.nn_ls2_26))) | 0.83 RMSE |
| Task 8: recognize_digit(1) | 1. (nn_ls2_41, lib.nn_ls2_3) | 0.29% CE |
| | 2. (nn_ls2_42, lib.nn_ls2_12) | 0.19% CE |
| | 3. (nn_ls2_43, lib.nn_ls2_22) | 0.24% CE |
| Task 9: count_digit(8) | 1. (nn_ls2_44, ((nn_ls2_45, lib.nn_ls2_31))) | 0.46 RMSE |
| | 2. (nn_ls2_46, ((nn_ls2_47, lib.nn_ls2_26))) | 0.45 RMSE |
| | 3. (nn_ls2_48, ((nn_ls2_49, lib.nn_ls2_3))) | 0.47 RMSE |

Table 9: Long Sequence 3(LS3).

| Task | Top 3 Programs | Error |
|---|---|---|
| Task 1: count_digit(9) | 1. (nn_ls3_1, ((nn_ls3_2), (nn_ls3_3)))<br>2. (nn_ls3_4, ((nn_ls3_5, nn_ls3_6)))<br>3. ((nn_ls3_7, (nn_ls3_8)), (nn_ls3_9)) | 0.46 RMSE<br>0.48 RMSE<br>0.55 RMSE |
| Task 2: count_digit(1) | 1. (lib.nn_ls3_1, ((nn_ls3_10, nn_ls3_11)))<br>2. (lib.nn_ls3_1, ((nn_ls3_12), (nn_ls3_13)))<br>3. ((lib.nn_ls3_1, (nn_ls3_14)), (nn_ls3_15)) | 0.63 RMSE<br>0.68 RMSE<br>0.63 RMSE |
| Task 3: recognize_toy(2) | 1. (nn_ls3_16, nn_ls3_17)<br>2. (nn_ls3_18, lib.nn_ls3_11)<br>3. (lib.nn_ls3_2, nn_ls3_19) | 8.19% CE<br>9.95% CE<br>14.00% CE |
| Task 4: recognize_digit(1) | 1. (nn_ls3_20, lib.nn_ls3_3)<br>2. (nn_ls3_21, lib.nn_ls3_17)<br>3. (nn_ls3_22, nn_ls3_23) | 0.38% CE<br>0.48% CE<br>0.24% CE |
| Task 5: count_digit(3) | 1. (lib.nn_ls3_1, ((nn_ls3_24, nn_ls3_25)))<br>2. (lib.nn_ls3_1, ((nn_ls3_26, lib.nn_ls3_17)))<br>3. (lib.nn_ls3_1, ((nn_ls3_27, lib.nn_ls3_11))) | 0.51 RMSE<br>0.66 RMSE<br>0.61 RMSE |
| Task 6: count_digit(1) | 1. (nn_ls3_28, ((nn_ls3_29, lib.nn_ls3_11)))<br>2. (nn_ls3_30, ((nn_ls3_31, lib.nn_ls3_3)))<br>3. (lib.nn_ls3_1, ((nn_ls3_32, lib.nn_ls3_3))) | 0.38 RMSE<br>0.37 RMSE<br>0.40 RMSE |
| Task 7: count_digit(2) | 1. (nn_ls3_33, ((nn_ls3_34, lib.nn_ls3_17)))<br>2. (lib.nn_ls3_1, ((nn_ls3_35, nn_ls3_36)))<br>3. (lib.nn_ls3_1, ((nn_ls3_37, lib.nn_ls3_17))) | 0.96 RMSE<br>0.99 RMSE<br>0.90 RMSE |
| Task 8: recognize_digit(9) | 1. (nn_ls3_38, nn_ls3_39)<br>2. (lib.nn_ls3_2, nn_ls3_40)<br>3. (lib.nn_ls3_2, lib.nn_ls3_3) | 1.52% CE<br>2.43% CE<br>1.43% CE |
| Task 9: count_digit(3) | 1. (nn_ls3_41, ((nn_ls3_42, nn_ls3_43)))<br>2. (nn_ls3_44, ((nn_ls3_45, lib.nn_ls3_39)))<br>3. (nn_ls3_46, ((nn_ls3_47, lib.nn_ls3_3))) | 0.39 RMSE<br>0.42 RMSE<br>0.44 RMSE |

Table 10: Long Sequence 4(LS4).

| Task | Top 3 Programs | Error |
|---|---|---|
| Task 1: count_digit(6) | 1. (nn_ls4_1, ((nn_ls4_2), (nn_ls4_3)))<br>2. (nn_ls4_4, ((nn_ls4_5, nn_ls4_6)))<br>3. ((nn_ls4_7, (nn_ls4_8)), (nn_ls4_9)) | 0.40 RMSE<br>0.45 RMSE<br>0.48 RMSE |
| Task 2: count_digit(2) | 1. (lib.nn_ls4_1, ((nn_ls4_10), (nn_ls4_11)))<br>2. (lib.nn_ls4_1, ((nn_ls4_12, nn_ls4_13)))<br>3. ((lib.nn_ls4_1, (nn_ls4_14)), (nn_ls4_15)) | 0.89 RMSE<br>0.99 RMSE<br>0.91 RMSE |
| Task 3: recognize_toy(3) | 1. (nn_ls4_16, lib.nn_ls4_11)<br>2. (nn_ls4_17, nn_ls4_18)<br>3. (lib.nn_ls4_10, nn_ls4_19) | 4.95% CE<br>4.00% CE<br>2.43% CE |
| Task 4: recognize_digit(8) | 1. (nn_ls4_20, lib.nn_ls4_3)<br>2. (nn_ls4_21, nn_ls4_22)<br>3. (nn_ls4_23, lib.nn_ls4_11) | 0.71% CE<br>0.52% CE<br>0.86% CE |
| Task 5: count_digit(1) | 1. (lib.nn_ls4_1, ((nn_ls4_24, nn_ls4_25)))<br>2. (lib.nn_ls4_1, ((nn_ls4_26, lib.nn_ls4_11)))<br>3. (lib.nn_ls4_1, ((lib.nn_ls4_16, nn_ls4_27))) | 0.64 RMSE<br>0.57 RMSE<br>0.70 RMSE |
| Task 6: count_digit(8) | 1. (lib.nn_ls4_1, ((nn_ls4_28, lib.nn_ls4_3)))<br>2. (lib.nn_ls4_1, ((nn_ls4_29, nn_ls4_30)))<br>3. (lib.nn_ls4_1, ((nn_ls4_31, lib.nn_ls4_11))) | 0.39 RMSE<br>0.38 RMSE<br>0.40 RMSE |
| Task 7: count_digit(3) | 1. (lib.nn_ls4_1, ((nn_ls4_32, lib.nn_ls4_11)))<br>2. (lib.nn_ls4_1, ((nn_ls4_33, nn_ls4_34)))<br>3. (lib.nn_ls4_1, ((nn_ls4_35, lib.nn_ls4_25))) | 0.61 RMSE<br>0.54 RMSE<br>0.60 RMSE |
| Task 8: recognize_digit(6) | 1. (nn_ls4_36, lib.nn_ls4_3)<br>2. (lib.nn_ls4_20, nn_ls4_37)<br>3. (nn_ls4_38, nn_ls4_39) | 0.81% CE<br>0.90% CE<br>0.86% CE |
| Task 9: count_digit(5) | 1. (lib.nn_ls4_1, ((nn_ls4_40, nn_ls4_41)))<br>2. (lib.nn_ls4_1, ((nn_ls4_42, lib.nn_ls4_3)))<br>3. (lib.nn_ls4_1, ((nn_ls4_43, lib.nn_ls4_11))) | 0.37 RMSE<br>0.39 RMSE<br>0.39 RMSE |

Table 11: Long Sequence 5(LS5).

| Task | Top 3 Programs | Error |
|------|----------------|-------|
| Task 1: count_digit(4) | 1. (nn_ls5_1, ((nn_ls5_2), (nn_ls5_3))) | 0.45 RMSE |
| | 2. (nn_ls5_4, ((nn_ls5_5, nn_ls5_6))) | 0.46 RMSE |
| | 3. ((nn_ls5_7, (nn_ls5_8)), (nn_ls5_9)) | 0.48 RMSE |
| Task 2: count_digit(3) | 1. (nn_ls5_10, ((nn_ls5_11), (lib.nn_ls5_3))) | 0.60 RMSE |
| | 2. (lib.nn_ls5_1, ((nn_ls5_12), (nn_ls5_13))) | 0.63 RMSE |
| | 3. ((lib.nn_ls5_1, (nn_ls5_14)), (nn_ls5_15)) | 0.58 RMSE |
| Task 3: recognize_toy(4) | 1. (nn_ls5_16, nn_ls5_17) | 20.33% CE |
| | 2. (lib.nn_ls5_11, nn_ls5_18) | 17.76% CE |
| | 3. (nn_ls5_19, lib.nn_ls5_3) | 21.38% CE |
| Task 4: recognize_digit(7) | 1. (nn_ls5_20, nn_ls5_21) | 1.19% CE |
| | 2. (nn_ls5_22, lib.nn_ls5_3) | 0.90% CE |
| | 3. (lib.nn_ls5_2, nn_ls5_23) | 1.62% CE |
| Task 5: count_digit(0) | 1. (lib.nn_ls5_10, ((nn_ls5_24, nn_ls5_25))) | 0.90 RMSE |
| | 2. (lib.nn_ls5_10, ((nn_ls5_26, lib.nn_ls5_17))) | 0.90 RMSE |
| | 3. (lib.nn_ls5_10, ((nn_ls5_27, lib.nn_ls5_3))) | 0.86 RMSE |
| Task 6: count_digit(7) | 1. (lib.nn_ls5_1, ((nn_ls5_28, nn_ls5_29))) | 0.47 RMSE |
| | 2. (lib.nn_ls5_1, ((nn_ls5_30, lib.nn_ls5_21))) | 0.47 RMSE |
| | 3. (nn_ls5_31, ((lib.nn_ls5_16, nn_ls5_32))) | 0.47 RMSE |
| Task 7: count_digit(4) | 1. (nn_ls5_33, ((nn_ls5_34, lib.nn_ls5_25))) | 1.72 RMSE |
| | 2. (nn_ls5_35, ((nn_ls5_36, lib.nn_ls5_17))) | 1.50 RMSE |
| | 3. (lib.nn_ls5_1, ((nn_ls5_37, nn_ls5_38))) | 1.80 RMSE |
| Task 8: recognize_digit(4) | 1. (nn_ls5_39, lib.nn_ls5_3) | 0.29% CE |
| | 2. (nn_ls5_40, lib.nn_ls5_21) | 0.38% CE |
| | 3. (lib.nn_ls5_20, nn_ls5_41) | 0.48% CE |
| Task 9: count_digit(0) | 1. (nn_ls5_42, ((nn_ls5_43, nn_ls5_44))) | 0.37 RMSE |
| | 2. (nn_ls5_45, ((lib.nn_ls5_24, nn_ls5_46))) | 0.40 RMSE |
| | 3. (nn_ls5_47, ((nn_ls5_48, lib.nn_ls5_21))) | 0.40 RMSE |

Table 12: Counting Sequence 1(CS1), Evolutionary Algorithm. "CE" denotes classification error and "RMSE" denotes root mean square error.

| Task | Top 3 programs | Error |
|------|----------------|-------|
| Task 1: recognize_digit($d_1$) | 1. (nn_cs1_1, nn_cs1_2) | 0.57% CE |
| | 2. (nn_cs1_3, nn_cs1_2) | 0.38% CE |
| | 3. (nn_cs1_4, nn_cs1_2) | 0.76% CE |
| Task 2: recognize_digit($d_2$) | 1. (nn_cs1_5, nn_cs1_6) | 0.38% CE |
| | 2. (nn_cs1_7, nn_cs1_8) | 0.48% CE |
| | 3. (nn_cs1_9, nn_cs1_10) | 0.43% CE |
| Task 3: count_digit($d_1$) | 1. (nn_cs1_11, ((nn_cs1_12, lib.nn_cs1_2))) | 0.38 RMSE |
| | 2. (nn_cs1_13, ((nn_cs1_14, lib.nn_cs1_2))) | 0.38 RMSE |
| | 3. (nn_cs1_15, ((lib.nn_cs1_1, lib.nn_cs1_2))) | 0.40 RMSE |
| Task 4: count_digit($d_2$) | No Solution | |

Table 13: Counting Sequence 2(CS2), Evolutionary Algorithm.

| Task | Top 3 programs | Error |
|------|----------------|-------|
| Task 1: recognize_digit($d_1$) | 1. (nn_cs2_1, nn_cs2_2) | 1% CE |
| | 2. (nn_cs2_3, nn_cs2_4) | 1% CE |
| | 3. (nn_cs2_5, nn_cs2_2) | 1% CE |
| Task 2: count_digit($d_1$) | 1. (nn_cs2_6, ((lib.nn_cs2_1, lib.nn_cs2_2))) | 0.38 RMSE |
| | 2. (nn_cs2_6, ((nn_cs2_7, lib.nn_cs2_2))) | 0.38 RMSE |
| | 3. (nn_cs2_8, ((nn_cs2_9, nn_cs2_10))) | 0.39 RMSE |
| Task 3: count_digit($d_2$) | No Solution | |
| Task 4: recognize_digit($d_2$) | 1. (nn_cs2_11, nn_cs2_12) | 1% CE |
| | 2. (nn_cs2_13, nn_cs2_14) | 1% CE |
| | 3. (lib.nn_cs2_1, nn_cs2_15) | 1% CE |

Table 14: Counting Sequence 3(CS3), Evolutionary Algorithm.

| Task | Top 3 Programs | Error |
|------|----------------|-------|
| Task 1: recognize_digit($d$) | 1. (nn_cs3_1, nn_cs3_2) | 0.57% CE |
| | 2. (nn_cs3_3, nn_cs3_4) | 0.67% CE |
| | 3. (nn_cs3_5, nn_cs3_6) | 0.62% CE |
| Task 2: count_digit($d$) | 1. (nn_cs3_7, ((nn_cs3_8, lib.nn_cs3_2))) | 0.36 RMSE |
| | 2. (nn_cs3_7, ((nn_cs3_9, nn_cs3_10))) | 0.39 RMSE |
| | 3. (nn_cs3_11, ((nn_cs3_12, nn_cs3_13))) | 0.39 RMSE |
| Task 3: count_toy($t$) | 1. (lib.nn_cs3_7, ((nn_cs3_14, nn_cs3_15))) | 0.70 RMSE |
| | 2. (lib.nn_cs3_7, ((nn_cs3_16, nn_cs3_17))) | 0.61 RMSE |
| | 3. (lib.nn_cs3_7, ((nn_cs3_18, nn_cs3_19))) | 0.64 RMSE |
| Task 4: recognize_toy($t$) | 1. (nn_cs3_20, lib.nn_cs3_15) | 5.62% CE |
| | 2. (lib.nn_cs3_14, lib.nn_cs3_15) | 5.38% CE |
| | 3. (nn_cs3_21, lib.nn_cs3_15) | 5.76% CE |

## Table 15: Summing Sequence(SS), Evolutionary Algorithm

| Task | Top 3 programs | Error |
|---|---|---|
| Task 1: classify_digit | 1. (nn_ss_1, nn_ss_2) | 1% CE |
| Task 2: sum_digits | 1. (( nn_ss_3 zeros(1)), ((nn_ss_4, lib.nn_ss_2))) | 6.64 RMSE |
| | 2. (( nn_ss_5 zeros(1)), ((nn_ss_6, lib.nn_ss_2))) | 6.66 RMSE |
| | 3. (( nn_ss_7 zeros(1)), ((nn_ss_8, lib.nn_ss_2))) | 6.70 RMSE |

## Table 16: Graph Sequence 1(GS1), Evolutionary Algorithm.

| Task | Top 3 Programs | Error |
|---|---|---|
| Task 1: regress_speed | 1. (nn_gs1_1, nn_gs1_2) | 0.80 RMSE |
| Task 2: shortest_path_street | 1. ((nn_gs1_3, lib.nn_gs1_2)) | 8.36 RMSE |
| | 2. ((nn_gs1_4, nn_gs1_5)) | 8.37 RMSE |
| | 3. ((nn_gs1_6, lib.nn_gs1_2)) | 8.35 RMSE |

## Table 17: Graph Sequence 2(GS2), Evolutionary Algorithm.

| Task | Top 3 Programs | Error |
|---|---|---|
| Task 1: regress_mnist | 1. (nn_gs2_1, nn_gs2_2) | 1.47 RMSE |
| Task 2: shortest_path_mnist | 1. ((lib.nn_gs2_1, nn_gs2_3)) | 6.58 RMSE |
| | 2. ((lib.nn_gs2_1, nn_gs2_4)) | 6.59 RMSE |
| | 3. ((lib.nn_gs2_1, nn_gs2_5)) | 6.63 RMSE |
| Task 3: shortest_path_street | 1. ((lib.nn_gs2_1, nn_gs2_6)) | 7.82 RMSE |
| | 2. ((lib.nn_gs2_1, nn_gs2_7)) | 7.87 RMSE |
| | 3. ((nn_gs2_8, nn_gs2_9)) | 7.96 RMSE |

## Table 18: Long Sequence 1(LS1), Evolutionary Algorithm.

| Task | Top 3 Programs | Error |
|---|---|---|
| Task 1: count_digit(7) | 1. (nn_ls1_1, ((nn_ls1_2, nn_ls1_3))) | 0.42 RMSE |
| | 2. (nn_ls1_4, ((nn_ls1_5, nn_ls1_6))) | 0.44 RMSE |
| | 3. (nn_ls1_1, ((nn_ls1_7, nn_ls1_8))) | 0.50 RMSE |
| Task 2: count_digit(4) | 1. (lib.nn_ls1_1, ((nn_ls1_9, nn_ls1_10))) | 1.65 RMSE |
| | 2. (lib.nn_ls1_1, ((nn_ls1_11, nn_ls1_12))) | 1.53 RMSE |
| | 3. (lib.nn_ls1_1, ((nn_ls1_13, nn_ls1_14))) | 1.60 RMSE |
| Task 3: recognize_toy(0) | 1. (nn_ls1_15, lib.nn_ls1_10) | 9.81% CE |
| | 2. (nn_ls1_16, nn_ls1_17) | 9.76% CE |
| | 3. (nn_ls1_18, lib.nn_ls1_10) | 8.76% CE |
| Task 4: recognize_digit(9) | 1. (nn_ls1_19, nn_ls1_20) | 1.43% CE |
| | 2. (nn_ls1_21, nn_ls1_22) | 1.43% CE |
| | 3. (nn_ls1_23, nn_ls1_24) | 1.62% CE |
| Task 5: count_digit(2) | 1. (nn_ls1_25, ((lib.nn_ls1_19, nn_ls1_26))) | 0.90 RMSE |
| | 2. (lib.nn_ls1_1, ((nn_ls1_27, nn_ls1_28))) | 0.94 RMSE |
| | 3. (lib.nn_ls1_1, ((nn_ls1_29, nn_ls1_30))) | 0.98 RMSE |
| Task 6: count_digit(9) | No Solution | |
| Task 7: count_digit(0) | No Solution | |
| Task 8: recognize_digit(7) | 1. (nn_ls1_31, lib.nn_ls1_26) | 1.29% CE |
| | 2. (nn_ls1_32, lib.nn_ls1_3) | 0.71% CE |
| | 3. (nn_ls1_33, nn_ls1_34) | 1.19% CE |
| Task 9: count_digit(2) | 1. (nn_ls1_35, ((nn_ls1_36, lib.nn_ls1_3))) | 0.36 RMSE |
| | 2. (lib.nn_ls1_25, ((nn_ls1_36, nn_ls1_37))) | 0.38 RMSE |
| | 3. (nn_ls1_38, ((nn_ls1_39, nn_ls1_40))) | 0.37 RMSE |

## Table 19: Long Sequence 2(LS2), Evolutionary Algorithm.

| Task | Top 3 Programs | Error |
|---|---|---|
| Task 1: count_digit(1) | No Solution | |
| Task 2: count_digit(0) | No Solution | |
| Task 3: recognize_toy(1) | 1. (nn_ls2_1, nn_ls2_2) | 5.43% CE |
| | 2. (nn_ls2_3, nn_ls2_4) | 5.81% CE |
| | 3. (nn_ls2_5, nn_ls2_6) | 5.05% CE |
| Task 4: recognize_digit(5) | 1. (nn_ls2_7, nn_ls2_8) | 0.71% CE |
| | 2. (nn_ls2_9, nn_ls2_10) | 0.43% CE |
| | 3. (nn_ls2_9, nn_ls2_11) | 0.62% CE |
| Task 5: count_digit(4) | No Solution | |
| Task 6: count_digit(5) | No Solution | |
| Task 7: count_digit(1) | No Solution | |
| Task 8: recognize_digit(1) | 1. (nn_ls2_12, nn_ls2_13) | 0.19% CE |
| | 2. (nn_ls2_14, lib.nn_ls2_2) | 0.29% CE |
| | 3. (nn_ls2_15, lib.nn_ls2_2) | 0.33% CE |
| Task 9: count_digit(8) | No Solution | |

Table 20: Long Sequence 3(LS3), Evolutionary Algorithm.

| Task | Top 3 Programs | Error |
|---|---|---|
| Task 1: count_digit(9) | No Solution | |
| Task 2: count_digit(1) | No Solution | |
| Task 3: recognize_toy(2) | 1. (nn_ls3_1, nn_ls3_2) | 10.52% CE |
| | 2. (nn_ls3_3, nn_ls3_2) | 9.14% CE |
| | 3. (nn_ls3_4, nn_ls3_2) | 10.81% CE |
| Task 4: recognize_digit(1) | 1. (nn_ls3_5, nn_ls3_6) | 0.48% CE |
| | 2. (nn_ls3_7, lib.nn_ls3_2) | 0.33% CE |
| | 3. (nn_ls3_8, nn_ls3_9) | 0.24% CE |
| Task 5: count_digit(3) | No Solution | |
| Task 6: count_digit(1) | 1. (nn_ls3_10, ((nn_ls3_11, lib.nn_ls3_6))) | 0.38 RMSE |
| | 2. (nn_ls3_12, ((nn_ls3_13, nn_ls3_14))) | 0.37 RMSE |
| | 3. (nn_ls3_15, ((nn_ls3_11, lib.nn_ls3_6))) | 0.39 RMSE |
| Task 7: count_digit(2) | 1. (lib.nn_ls3_10, ((nn_ls3_16, nn_ls3_17))) | 1.02 RMSE |
| | 2. (lib.nn_ls3_10, ((nn_ls3_18, nn_ls3_17))) | 0.92 RMSE |
| | 3. (lib.nn_ls3_10, ((nn_ls3_16, nn_ls3_19))) | 0.96 RMSE |
| Task 8: recognize_digit(9) | 1. (nn_ls3_20, nn_ls3_21) | 1.05% CE |
| | 2. (nn_ls3_22, nn_ls3_23) | 1.14% CE |
| | 3. (nn_ls3_24, lib.nn_ls3_17) | 1.86% CE |
| Task 9: count_digit(3) | 1. (lib.nn_ls3_10, ((nn_ls3_25, lib.nn_ls3_21))) | 0.45 RMSE |
| | 2. (lib.nn_ls3_10, ((nn_ls3_26, lib.nn_ls3_17))) | 0.47 RMSE |
| | 3. (lib.nn_ls3_10, ((nn_ls3_25, lib.nn_ls3_21))) | 0.46 RMSE |

Table 21: Long Sequence 4(LS4), Evolutionary Algorithm.

| Task | Top 3 Programs | Error |
|---|---|---|
| Task 1: count_digit(6) | No Solution | |
| Task 2: count_digit(2) | No Solution | |
| Task 3: recognize_toy(3) | 1. (nn_ls4_1, nn_ls4_2) | 5.10% CE |
| | 2. (nn_ls4_3, nn_ls4_4) | 3.57% CE |
| | 3. (nn_ls4_5, nn_ls4_6) | 4.24% CE |
| Task 4: recognize_digit(8) | 1. (nn_ls4_7, nn_ls4_8) | 0.33% CE |
| | 2. (nn_ls4_9, nn_ls4_10) | 0.48% CE |
| | 3. (nn_ls4_11, nn_ls4_12) | 0.90% CE |
| Task 5: count_digit(1) | No Solution | |
| Task 6: count_digit(8) | 1. (nn_ls4_13, ((nn_ls4_14, nn_ls4_15))) | 0.41 RMSE |
| | 2. (nn_ls4_16, ((nn_ls4_17, lib.nn_ls4_8))) | 0.44 RMSE |
| | 3. (nn_ls4_18, ((nn_ls4_19, lib.nn_ls4_8))) | 0.44 RMSE |
| Task 7: count_digit(3) | 1. (lib.nn_ls4_13, ((nn_ls4_20, nn_ls4_21))) | 0.56 RMSE |
| | 2. (lib.nn_ls4_13, ((nn_ls4_22, lib.nn_ls4_2))) | 0.59 RMSE |
| | 3. (lib.nn_ls4_13, ((nn_ls4_20, lib.nn_ls4_2))) | 0.57 RMSE |
| Task 8: recognize_digit(6) | 1. (nn_ls4_23, lib.nn_ls4_15) | 0.48% CE |
| | 2. (nn_ls4_24, lib.nn_ls4_15) | 0.62% CE |
| | 3. (nn_ls4_25, lib.nn_ls4_15) | 0.71% CE |
| Task 9: count_digit(5) | 1. (lib.nn_ls4_13, ((nn_ls4_26, nn_ls4_27))) | 0.41 RMSE |
| | 2. (lib.nn_ls4_13, ((nn_ls4_26, nn_ls4_27))) | 0.41 RMSE |
| | 3. (lib.nn_ls4_13, ((nn_ls4_28, nn_ls4_29))) | 0.41 RMSE |

Table 22: Long Sequence 5(LS5), Evolutionary Algorithm.

| Task | Top 3 Programs | Error |
|---|---|---|
| Task 1: count_digit(4) | No Solution | |
| Task 2: count_digit(3) | No Solution | |
| Task 3: recognize_toy(4) | 1. (nn_ls5_1, nn_ls5_2) | 17.00% CE |
| | 2. (nn_ls5_3, nn_ls5_4) | 21.62% CE |
| | 3. (nn_ls5_5, nn_ls5_6) | 16.52% CE |
| Task 4: recognize_digit(7) | 1. (nn_ls5_7, nn_ls5_8) | 1.14% CE |
| | 2. (nn_ls5_9, nn_ls5_10) | 0.95% CE |
| | 3. (nn_ls5_11, nn_ls5_12) | 1.00% CE |
| Task 5: count_digit(0) | No Solution | |
| Task 6: count_digit(7) | No Solution | |
| Task 7: count_digit(4) | No Solution | |
| Task 8: recognize_digit(4) | 1. (nn_ls5_13, nn_ls5_14) | 0.38% CE |
| | 2. (nn_ls5_15, lib.nn_ls5_8) | 0.33% CE |
| | 3. (nn_ls5_15, nn_ls5_16) | 0.33% CE |
| Task 9: count_digit(0) | 1. (nn_ls5_17, ((nn_ls5_18, lib.nn_ls5_8))) | 0.38 RMSE |
| | 2. (nn_ls5_17, ((nn_ls5_19, lib.nn_ls5_2))) | 0.38 RMSE |
| | 3. (nn_ls5_17, ((nn_ls5_20, lib.nn_ls5_8))) | 0.40 RMSE |

(a) Task 1: recognize_digit($d_1$)

(b) Task 2: recognize_digit($d_2$)

(c) Task 3: count_digit($d_1$)

(d) Task 4: count_digit($d_2$)

Figure 2: Lifelong learning for "learning to count" (Sequence CS1), demonstrating low-level transfer of perceptual recognizers.

# G  Results on Longer Task Sequence LS

We report the performance of all methods on the longer task sequences on Figure 5. To save space, we report performance of all methods when trained on 10% of the data. The full learning curves follow similar patterns as the other task sequences. We report the classification and regression tasks from LS separately, because the error functions for the two tasks have different dynamic ranges. Please note that in the Figure, the tasks are labelled starting from 0. On the classification tasks, we note that all methods have similar performance. Examining the task sequence LS from Figure **??**, we see that these tasks have no opportunity to transfer from earlier tasks. On the regression tasks however, there is opportunity to transfer, and we see that HOUDINI shows much better performance than the other methods.

(a) Task 1: recognize_digit($d_1$)

(b) Task 2: count_digit($d_1$)

(c) Task 3: count_digit($d_2$)

(d) Task 4: recognize_digit($d_2$)

Figure 3: Lifelong learning for "learning to count" (Sequence CS2), demonstrating high-level transfer of a counting network across categories.

(a) Task 1: recognize_digit($d_1$)

(b) Task 2: count_digit($d_1$)

(c) Task 3: count_toy($t_1$)

(d) Task 4: recognize_toy($t_1$)

Figure 4: Lifelong learning for "learning to count" (Sequence CS3), demonstrating high-level transfer across different types of images. After learning to count MNIST digits, the same network can be used to count images of toys.

Figure 5: Performance of transfer learning systems on task sequence LS1. At top: regression tasks. At bottom: classification tasks