[Reviews · NeurIPS 2018]

Reviewer 1



The authors present an algorithm for transfer learning using a symbolic program synthesizer for finding the most adequate neural network architecture and selecting relevant neural network modules from previous tasks for transfer. The approach is heavily based on concepts from programming languages, but also studies the relevant concept of high-level transfer that is crucial for true lifelong learning. Results show how the algorithm is capable of selectively transferring (high- and low-level) knowledge in a meaningful way, and numerical results validate the significance of the approach. The authors claim that their method targets the lifelong learning problem, but theirs is really a transfer learning approach. Solving catastrophic forgetting by completely freezing the network parameters precludes the method from being true lifelong learning, in which the learning of subsequent tasks affects the performance of earlier tasks. As a transfer learning method, it is quite interesting in that it selects which parts of the network of a previous task are useful to transfer over to a new task and with which combinator, allowing it to create the relevant transfer architecture via search. This is apparently the first study of program synthesis for differentiable programs. The idea of using program search for transfer learning is novel, and provides an interesting avenue for future work, although the search methods themselves don't seem novel. The usage of type-based pruning dramatically reduces the search-space, which is the only way for this approach to be applicable in a realistic learning scenario. The presentation is very complete, with a good amount of details on the proposed language and clear explanations on how these details are useful (and necessary) for learning. However, other details like the specific heuristic used for sorting the program search queue should be provided. As for the learning algorithm, it seems unjustified to simply compare validation performance as a selection criterion for the network architecture, especially since in a transfer learning context it is many times infeasible to set aside a validation set. Since the approach is presented as solving an optimization problem, using the cost function as the selection criterion seems better. The experimental setting was good, and discussing (and showing in the appendix) the actual programs learned by the approach was helpful to see the actual capabilities of the algorithm of transferring knowledge. I would have liked to see some randomized experiments where the sequences of tasks were not chosen specifically to highlight the algorithm's strengths, although it was good that many different sequences were used. It is unclear to me what dataset size was used for the actual programs learned by the method showed in the appendix; this should be specified. I notice a couple of presentation issues. First, Figure 3 is actually a table, and should be presented as such. Second, all the appendices are referenced as Appendix, but it would be far more helpful if the actual appendix number were used. Figure 4 is presented almost 2 pages after it is first discussed, which makes it awkward to look for. Typos: - Line 101: "Figure 1 gives shows..." - Line 274: "Leanrning a effective..."

Reviewer 2



This paper discusses transfer learning for program synthesis from input-output examples. The paper presents a new language to define the space of differentiable functional programs, which combines differentiable neural network layers, as well as non-differentiable functional constructors, i.e., maps and folds. Each program in the language can be trained as a neural network, which transforms inputs into outputs. The paper also presents two heuristics for sampling programs from the space. Some evaluation results are presented. In general, the main issue is that it is unclear what is the main contribution of this work. The language is new, though simple. The sampling algorithm can actually be seen as an architecture search algorithm, but not discussed in depth. The hardness of handling branching is not discussed in this work, since not needed; as a result, training a program, which is essentially a neural network architecture, is pretty standard. So far, nothing very surprising. I was surprised by the claim that the approach can handle shortest path algorithms. However, the results are presented only in the supplemental materials. By digging into the results, the RMSE can be as large as 5.51, which almost means a failure for a small graph. I would suggest the authors better organize the materials for the next version. In particular, I observe that the authors are facing the challenge from the space requirements, but the paper can be better organized as follows. (1) Fig 3 can be replaced with an end-to-end example to show the interesting aspects of the proposed algorithm. Right now, an end-to-end example is missing, which makes understanding this work very hard. (2) The shortest-path example, which is the most compelling application of this work, should be presented in full in the main body. Other less important and sometime tedious details can be deferred to the appendix. (3) The introduction should have a short paragraph summarizing a list of the most important contributions. Some minor ones: for the syntax in Fig 1, Atom is a TT, which is a ADT, which is a F, which is in \tau. Thus, it can be defined \tau = ADT | \tau->\tau and removing F without hurting the correctness. This will also saves you a line.

Reviewer 3



This paper proposed using symbolic program synthesis to solve the lifelong learning problem: a symbolic synthesizer dose type-directed search over a library of previously trained neural functions to generate a model for a given tasks, then a neural module fine tunes this generated model using SGD. This method called HOUDINI mainly addresses 2 problems in lifelong learning: (1) catastrophic forgetting is prevented by the use of function library in which the trained models are frozen; (2) negative transfer is prevented because HOUDINI is shown to be able to navigate itself to the right model for reuse. The strength of HOUDINI rely on 2 things: First, the models in the library are differentiable, this use of differentiable programing language technique is not new (the authors have mentioned the existing approach NTPT), but they advance it by developing a new typed functional language also called HOUDINI to combine these models. Second, such language also makes it possible to represent a wide range of architecture (CNN, RNN, etc) in a compact way and speed up the search (by reducing the search space to only type-safe modules). HOUDINI has been validated using tasks with different level of transfer. The experiment results show that HOUDINI has better initialization and stays outperformed than other considered methods. (1) Quality and clarity: The paper is generally clear. The problem is explained, experiments are designed to consider different transfer situations. But the authors should improve the structure to make the paper more clear. After reading the authors' responses, I found some key information is absent in this first version. Having an end-to-end example and explain it well will help readers understand questions like why ‘nn_cs1_2’ is a correct learning source for count_d1 task. Also, the authors should comment more on their experiment results, for example the ‘jumps’ around training dataset size ~ 3000 (in the authors response they promised to detail this), (2) Originality: The proposed language HOUDINI is quite original thanks to its combination of functional language and symbolic synthesis. Also using types to reduce research is quite creative.